# Rhizobial Exopolysaccharides: Genetic Regulation of Their Synthesis and Relevance in Symbiosis with Legumes

**DOI:** 10.3390/ijms22126233

**Published:** 2021-06-09

**Authors:** Sebastián Acosta-Jurado, Francisco Fuentes-Romero, Jose-Enrique Ruiz-Sainz, Monika Janczarek, José-María Vinardell

**Affiliations:** 1Department of Microbiology, University of Sevilla, Avda. Reina Mercedes 6, 41012 Seville, Spain; sacosta@us.es (S.A.-J.); ffuentesr@us.es (F.F.-R.); rsainz@us.es (J.-E.R.-S.); 2Department of Industrial and Environmental Microbiology, Institute of Biological Sciences, Faculty of Biology and Biotechnology, Maria Curie-Skłodowska University, Akademicka 19, 20-033 Lublin, Poland

**Keywords:** rhizobia, legume, rhizobium–legume symbiosis, exopolysaccharide synthesis, quorum sensing (QS), nodulation, *nod* regulon, flavonoids, RosR/MucR, SyrM

## Abstract

Rhizobia are soil proteobacteria able to engage in a nitrogen-fixing symbiotic interaction with legumes that involves the rhizobial infection of roots and the bacterial invasion of new organs formed by the plant in response to the presence of appropriate bacterial partners. This interaction relies on a complex molecular dialogue between both symbionts. Bacterial *N*-acetyl-glucosamine oligomers called Nod factors are indispensable in most cases for early steps of the symbiotic interaction. In addition, different rhizobial surface polysaccharides, such as exopolysaccharides (EPS), may also be symbiotically relevant. EPS are acidic polysaccharides located out of the cell with little or no cell association that carry out important roles both in free-life and in symbiosis. EPS production is very complexly modulated and, frequently, co-regulated with Nod factors, but the type of co-regulation varies depending on the rhizobial strain. Many studies point out a signalling role for EPS-derived oligosaccharides in root infection and nodule invasion but, in certain symbiotic couples, EPS can be dispensable for a successful interaction. In summary, the complex regulation of the production of rhizobial EPS varies in different rhizobia, and the relevance of this polysaccharide in symbiosis with legumes depends on the specific interacting couple.

## 1. Introduction

Rhizobia is the common name for a group of soil proteobacteria able to enter into symbiosis with legume plants [1]. As part of this interaction, rhizobia infect legume roots, the plant forms new organs (root nodules), and these organs are invaded by rhizobia: all together, these events are referred as the nodulation process. Inside nodules, rhizobia are hosted by symbiotic plant cells, in which they differentiate into bacteroids able to fix N_2_ into ammonia that, mostly, is assimilated by the plant [2,3]. Thanks to this symbiosis, legumes can grow in soils poor in nitrogen, which is important from both an ecological and an agricultural point of view. The inoculation of legumes with appropriate rhizobia can reduce or even eliminate the necessity of the application of nitrogen fertilizers (expensive and highly polluting), which makes the use of rhizobial inoculants very appropriate for sustainable agricultural practices [4,5].

The rhizobia–legume symbiotic interaction is characterized by being specific: each rhizobial strain can nodulate with a definite set of legumes (known as the nodulation range), which can be narrow or broad, and each legume species can be nodulated by a set of rhizobial strains [6]. This specificity relies on a complex molecular dialogue that is established between both symbionts [6,7,8,9]. Although there are exceptions [1,10], in most of the rhizobia–legume symbiotic interactions this dialogue can be summarized as follows [9,10]: plant root exudates contain different nutrients that can feed rhizobia, including flavonoids, which, when appropriate and through the activation of the bacterial LysR regulator NodD, trigger the expression of bacterial nodulation genes and thus, the production and secretion of a set of different Nod factors. These rhizobial molecules are N-acetyl-glucosamine oligosaccharides harbouring different chemical decorations that, when appropriate, are perceived by LysM receptors present in root hair membranes, eliciting different responses necessary for bacterial infection and the start of the nodule organogenesis program. The most evolved rhizobial infection pathway is through infection threads (IT): briefly, following cytoskeletal reorganization, root hairs curve and trap rhizobial cells that previously adhered to their tips. These bacteria become enclosed in a kind of pocket wherein the action of lytic enzymes produced by both symbionts leads to local plant cell-wall degradation. At that point, root hair membrane invaginates and originates a channel, the IT, that will be used by rhizobia to penetrate into the root. These channels grow towards the root cortex, where, in parallel to the infection process, nodule organogenesis had started. Several cortical cells dedifferentiated into meristematic cells and gave rise to the formation of the nodule primordium. Later, certain cells inside the nodule differentiate into high-size, symbiotic cells through a process of endoreplication (replication of the genome in the absence of cell division). IT, which have continued to grow and have also branched out, eventually enter into the forming nodule and liberate rhizobia into the nodule endoreduplicated cells by endocytosis. Inside these cells, rhizobia differentiate into bacteroids, able to fix nitrogen.

In addition to Nod factors, other rhizobial molecules play important roles in symbiotic interaction with legumes [7,8]. On the one hand, some rhizobia, such as *S. fredii* or *B. japonicum*, deliver effector proteins into plant cells through a type 3 secretion system [11]. These effector proteins mainly alter host signalling and suppress plant defences. On the other hand, different rhizobial surface polysaccharides participate in the nodulation process, most probably acting as signals required for the progression of the interaction and/or preventing plant defence mechanisms [7,8,12,13]. Rhizobia are Gram-negative bacteria, so they have an inner membrane (cytoplasmic membrane) and an outer membrane, separated by the periplasmic space that also contains the peptidoglycan layer. The main rhizobial surface polysaccharides studied so far are cyclic glucans (CG), lipopolysaccharides (LPS), capsular polysaccharides (CPS), K-antigen capsular polysaccharides (KPS) and exopolysaccharides (EPS).

CG are cyclic homopolymers of glucose residues that can harbour different substitutions such as 1-phosphoglycerol or succinate. CG are mainly present in the periplasmic space but can also be secreted to the extracellular milieu. LPS are very complex glycolipid molecules located in the outer leaflet of the outer membrane of Gram-negative bacteria. KPS are acidic polysaccharides analogous to the group II of K-antigens described in *Escherichia coli* that, among rhizobia, are present in the *Sinorhizobium* (=*Ensifer*) genus and located, as LPS, in the outer leaflet of the outer membrane. In *Rhizobium leguminosarum*, the term “capsular polysaccharide” (CPS) refers to different polysaccharides, either neutral or acidic, that form a matrix surrounding the bacteria [14]. Finally, EPS are acidic polysaccharides secreted to the cellular environment and located on the cell surface. The symbiotic relevance of each of these polysaccharides notably varies depending on the symbiotic couple analysed, with the only exception of CG, which have been shown to be essential in all the rhizobia–legume interactions studied so far [8,15].

There are other rhizobial surface polysaccharides, and some of them may play a role in symbiosis, reviewed by [7]. In *Rhizobium leguminosarum*, glucomannan (a water-soluble surface heteropolymer, consisting of 95% of glucose and mannose and small amounts of galactose and rhamnose), also known as neutral polysaccharide, is essential for the attachment of this bacterium to a lectin present in *Vicia sativa* and *Pisum sativum* root hairs. Also in *R. leguminosarum*, a neutral gel-forming polysaccharide containing galactose, mannose, and glucose in a molar ratio 4:1:1 is formed in in vitro cultures at the late stationary phase, but, apparently, it does not play a role in symbiosis. Finally, cellulose fibrils produced by rhizobia may participate in bacterial attachment to host root hairs and biofilm formation.

## 2. A General View of Rhizobial Exopolysaccharides

As mentioned above, EPS are acidic polysaccharides located out of the cell with little or no cell association [8]. These polysaccharides perform different functions, such as protection against environmental stresses, attachment to biotic and abiotic surfaces, and nutrient gathering [7,16]. During interactions with plants, diverse functions have been proposed for EPS from both pathogenic and symbiotic bacteria reviewed by [13], such as the sequestration of calcium ions, ROS (reactive oxygen species) scavenging, prevention of cellulose-mediated cell agglutination, tolerance to acidic pH, participation in biofilm formation, and host surface attachment. In addition, there is evidence of the role of EPS as a key signal for rhizobial infection in several rhizobia–legume couples as *Rhizobium leguminosarum* bv. *trifolii*/*Trifolium* spp., *Sinorhizobium meliloti*/*Medicago sativa* and *Mesorhizobium loti*/*Lotus japonicus* [7,16,17,18,19].

Rhizobial EPS are complex and high-molecular-mass heteropolymers whose structure varies at the species level. The repeating units are constituted of a variable number of hexose and uronic acid residues showing either alpha or beta glycosidic linkages. They can also be either linear or side-branched. In addition to sugars, several non-carbohydrate substituents can appear, mostly succinate, pyruvate, or acetate, which contribute to the acidic character of this polysaccharide [8,16,20]. Figure 1 shows the structure of the rhizobial EPS that will be treated in detail in this review. As it will be mentioned later, rhizobial EPS can be produced as either high- or low-molecular-mass forms (HMM and LMM).

Traditionally, the symbiotic relevance of rhizobial EPS has been related to the type of nodule ontogeny [16,17]. Thus, this polysaccharide appeared to be essential for indeterminate nodule-forming symbiosis, such as *S. meliloti*/*M. sativa* or *R. leguminosarum* bv. *trifolii*/*Trifolium* spp., but dispensable for determinate nodule-forming symbiosis, such as *Sinorhizobium fredii*/*Glycine max* and *M. loti*/*L. japonicus*. According to Stacey et al. [22] this difference may be due to the fact that the IT of indeterminate nodules are narrower than those of determinate nodules, and that the EPS-mediated matrix could be more important in the former ones.

In this review, we summarize the current knowledge about rhizobial EPS, focusing on the complex regulation of its production and on the symbiotic relevance of this polysaccharide. For the sake of clarity, we will restrict this review to four rhizobial species where EPS has been studied in more detail: *R. leguminosarum*, *S. meliloti*, *S. fredii*, and *M. loti.*
Appendix A lists all of the genes (including regulatory ones) involved in EPS production that are mentioned in this work. These studies will show that the symbiotic importance of rhizobial EPS depends more on the specific couple analysed than on the type of nodule, determinate or indeterminate, formed by the host legume.

## 3. *Rhizobium leguminosarum* 

*Rhizobium leguminosarum* is a rhizobial species in which three biovars, *trifolii*, *viciae* and *phaseoli*, can be distinguished based on the type of legumes infected. Each of these biovars exhibit a narrow host-range. *R. leguminosarum* bv. *trifolii* strains establish symbiosis with *Trifolium*, bv. *viciae* with *Pisum*, *Vicia*, *Lens*, and *Lathyrus*, and bv. *phaseoli* with *Phaseolus* spp. plants. The symbiotic importance of EPS in *R. leguminosarum* has been related to the type of nodules formed by these host plants [6,7,23,24]. *Trifolium*, *Pisum*, *Vicia*, *Lathyrus*, *Lens*, belonging to the galegoid clade of subfamily *Papilionoideae*, form indeterminate nodules, whereas determinate nodules are formed by phaseolid legumes such as *Phaseolus*. It is generally accepted that, in general, the production of EPS is required for the effective symbiosis of *R. leguminosarum trifolii* and *viciae* strains with their host plants, which form indeterminate nodules, but not for bv. *phaseoli* strains, which establish symbiosis with bean plants forming determinate nodules.

Despite these differences in the symbiotic importance of EPS, *R. leguminosarum* strains produce EPS showing similar, but not identical, structures [16] In most *R. leguminosarum* bv. *trifolii* strains (such as Rt42.2, RBL5599, LPR5, etc.), this polymer consists of octasaccharide repeating units which contain D-glucose, D-glucuronic acid and D-galactose in a molar ratio 5:2:1, joined by β-1,3 and β-1,4 glycosidic bonds, and are modified by non-sugar (acetyl and pyruvyl) groups (Figure 1), but in other strains (such as 4S), the galactose residue is not present [16,25,26,27,28]. In *R. leguminosarum* bv. *viciae*, the EPS repeating unit is very similar to the *R. leguminosarum* bv. *trifolii* octasaccharide but including an additional D-glucuronic acid residue [16,28].

Similarly to other microorganisms, EPS synthesis in rhizobia is a complex process requiring the coordinated activity of many enzymatic and transporter proteins [29]. Genes involved in the synthesis and secretion of rhizobial EPS are usually grouped in large clusters located on chromosomes or megaplasmids [30,31,32,33]. In the case of *R. leguminosarum*, a great majority of genes engaged in EPS production are located on a large chromosomal region, called Pss-I [34,35,36,37], that encompasses nearly 30 genes. This region is highly conserved among all the so far sequenced *R. leguminosarum* genomes, although differences in the genetic organization, due to rearrangements, can be found at the strain level [31,34,35,38,39,40]. Figure 2 shows the Pss-I region from *R. leguminosarum* bv. *trifolii* Rt24.2. This clustering of genes involved in EPS synthesis most probably reflects their coordinated regulation of expression, including the influence of various environmental factors (light, root exudates, nitrogen and phosphate limitation, catabolite repression, stress conditions) [41,42,43,44,45,46]. The Pss-I region contains the *exo5* gene, responsible for the synthesis of glucuronic acid (GlcA), genes encoding glycosyl transferases responsible for EPS subunit synthesis (*pssCDE*, *pssJIFGH*, *pssS*), genes involved in the addition of non-sugar modifications to EPS subunits (*pssRMK*), and genes involved in EPS assembly and secretion (*pssTNOP*, *pssL*) (Figure 2). The *exo5* gene codes for a UDP-glucose dehydrogenase which converts UDP-glucose to UDP-GlcA, which, in turns, is the source for the synthesis of galacturonic acid (GalA). A mutant in this gene displays pleiotropic effects (i.e., changes in bacterial cell envelope and ineffective symbiosis), because its inability to synthesize uronic acids results in the absence of EPS and capsular polysaccharide (CPS) production and in the synthesis of an altered LPS that lacks GalA residues [14,47].

Some chromosomal genes crucial for EPS production, such as *exoB* and *pssA*, are located outside the Pss-I region. The *exoB* gene codes for a UDP-glucose 4-epimerase that synthesizes UDP-galactose, being a donor of this sugar in the synthesis of EPS and other galactose-containing polysaccharides [51,52]. An *exoB* mutant produces EPS lacking galactose residues and is unable to properly invade host roots (abnormal nodules inefficient in nitrogen fixation were formed on clover roots). Figure 3 shows a putative model for the different steps of EPS subunit assembly and export in *R. leguminosarum* bv. *trifolii* strains in which this subunit is an octasaccharide containing D-glucose, D-glucuronic acid and D-galactose in a molar ratio 5:2:1. The *pssA* gene is involved in the first step of the EPS subunit assembly. This gene encodes a glucosyl-isoprenylphosphate transferase that transfers glucose-1-phosphate from UDP-glucose to the lipid carrier [53]. Mutations in *pssA* totally abolish EPS synthesis, leading to the induction of empty (without bacteria), non-nitrogen-fixing root nodules on host plants (clover, pea and vetch) [44,54,55,56,57,58]. The transcription of *pssA* in both *ex planta* and *in planta* conditions is at a low level, indicating that the expression of this gene, which is crucial for EPS synthesis, is under very stringent regulation [44,59,60,61].

All *pss* genes involved in subsequent steps of the EPS subunit synthesis belong to the Pss-I region. The addition of two glucuronosyl residues to the growing subunit is performed by a glucuronosyl-β-1,4-glucosyltransferase and a glucuronosyl-β-1,4-glucuronosyltransferase encoded by *pssDE* and *pssC*, respectively [36,53,55,62]. Interestingly, a *pssD* mutant shows very similar phenotypic traits as the *pssA* mutant (lack of EPS synthesis and non-nitrogen-fixing nodules induced on clover and vetch roots). The fourth step of the unit assembly is most probably conducted by PssS, which is a unique enzyme among *R. leguminosarum* glycosyltransferases engaged in the EPS synthesis because it generates an exclusive α-1,4-glucosyl bound [35]. However, there is little information about glycosyl transferases involved in the subsequent steps of EPS synthesis. Based on the CAZy database (Carbohydrate Active Enzymes), some *pss* genes located in the Pss-I region are predicted to be engaged in this process (*pssF*, *pssG*, *pssH*, *pssI*) [63,64]. As recently evidenced, *pssJ* codes for a galactosyl transferase that is involved in the addition of the terminal sugar residue to the octasaccharide subunits [65]. A Δ*pssJ* deletion mutant produces EPS lacking terminal galactose in the side chain of the subunit (Figure 1). However, the lack of addition of this galactose residue does not block EPS polymerization.

Three *pss* genes from the Pss-I region are involved in non-sugar modifications of the EPS structure (Figure 3); *pssR* codes for a putative acetyl transferase, whereas *pssM* and *pssK* code for ketal pyruvate transferases [66]. Mutation of *pssM* resulted in the absence of pyruvyl groups in the EPS subunits (at the sub-terminal glucose) (Figure 1) and the induction of non-nitrogen-fixing nodules on the host plant: in these nodules, the normal invasion and release of bacteria into plant cells were observed, but their differentiation into bacteroids was essentially impaired, revealing the essentiality of the addition of pyruvyl residues for the symbiotic function of this polysaccharide.

The polymerization and secretion of EPS in *R. leguminosarum* are carried out by proteins encoded by the *pssL* and *pssTNOP* genes from the Pss-I cluster (Figure 3). Detailed information about these processes can be found in recent reviews [29,64]. Briefly, the polymerization of EPS subunits is performed by a Wzx/Wzy system that is composed of a Wzx-type translocase (flippase), a Wzy-type polysaccharide polymerase and a co-polymerase. In *R. leguminosarum*, PssL is most probably a Wzx translocase involved in the transfer of the EPS subunits across the inner membrane. This type of polysaccharide transporter proteins (Wzx translocases) is characterized by a specific topology containing numerous (10–14) predicted transmembrane (TM) segments. PssT, a Wzy-type protein containing 12 TMs, is the polysaccharide polymerase responsible for the polymerization of the EPS subunits [67]. PssP is another component of this system (PCP, polysaccharide co-polymerase) that is involved in the determination of the EPS chains length [68]. This protein has been classified as a bacterial tyrosine kinase [69,70,71]. PssP displays a significant identity to *S. meliloti* ExoP and other membrane-periplasmic auxiliary (OMA) proteins that are involved in the synthesis of HMM surface polysaccharides. Transport of EPS outside bacterial cells is conducted through a channel in the outer membrane formed by Wza-type proteins that are lipoproteins forming octameric α-helical channels spanning both inner and outer membranes [72]. In *R. leguminosarum*, PssN plays the role of this outer membrane translocase that is able to form homooligomeric structures and interact with PssP [73].

Other genes related to EPS synthesis are also located in the Pss-I region, such as *prsDE* (coding for components of a type 1 secretion system) and *plyA* (coding for a glycanase that cleaves EPS, affecting its processing) [35,62,74]. This secretion system shows an atypical broad substrate specificity, exporting at least 13 protein substrates (e.g., glycanases (PlyA, PlyB, and PlyC), rhizobial adhesion proteins (RapA2, RapB, and RapC), and the nodulation protein NodO) [75,76,77,78]. A *prsD* mutant synthesizes EPS of a higher polymerization degree than that of the wild-type strain and elicits a higher number of nodules incapable of fixing nitrogen [75].

Recently, the role in EPS synthesis of another gene belonging to the Pss-I region, *pssZ*, has been shown (Figure 2). This gene codes for a protein belonging to the bacterial serine/threonine protein phosphatases family, and its inactivation totally abolished EPS production in *R. leguminosarum* bv. *trifolii* [79]. Comparative transcriptomic analysis of the *pssZ* mutant and the wild-type strain identified a large number of genes differentially expressed in these two backgrounds, including genes related to several cellular processes (signalling, transcription regulation, synthesis of cell-surface polysaccharides, cell division and motility, bacterial metabolism) [61]. Interestingly, expression of almost all *pss* genes located in the Pss-I region was totally inhibited in the *pssZ* mutant. These data confirmed the important role of the serine/threonine protein phosphatase PssZ in EPS synthesis, most probably via its participation in a regulatory network [71]. In bacteria, in addition to two-component signalling systems, alternative regulatory pathways controlled by Hanks-type serine/threonine kinases and serine/threonine phosphatases also play an essential role in regulation of many processes, such as growth and cell division, cell-wall biogenesis, sporulation, biofilm formation, stress response, and metabolic and developmental processes, as well as either symbiotic or pathogenic interactions with higher host organisms [70,80]. In fact, the *pssZ* mutation resulted in pleiotropic effects in rhizobial cells: a lack of EPS production, decreased growth kinetics and motility, altered cell-surface properties, and failure to infect the host plant. 

The global regulator RosR is involved in the positive regulation of EPS synthesis in *R. leguminosarum*. This protein is encoded by the chromosomal *rosR* gene that is not linked to the Pss-I region [81,82] and that is present in the genomes of all strains belonging to the three *R. leguminosarum* biovars and the closely related species *R. etli* and *R. gallicum* [83,84]. *R. leguminosarum rosR* also exhibits a significant similarity with *A. tumefaciens ros* [85], *mucR* of *S. meliloti* [86] and *S. fredii* [87,88], as well as with *mucR* of pathogenic *Brucella abortus* [89]. All these genes code for transcriptional regulators belonging to the α-proteobacteria Ros/MucR family of zinc-finger proteins which regulate the expression of genes required for the successful interactions of these bacteria with eukaryotic hosts. The *R. leguminosarum* bv. *trifolii rosR* mutant produces substantially less EPS (including a decreased LMM to HMM ratio) than the wild-type strain and establishes impaired symbiosis with clover plants [81,90]. In addition, alterations in the polysaccharide constituent of LPS, changes in membrane and secreted proteins, and decreased motility were observed in the *rosR* mutant [90,91]. On the other hand, additional *rosR* copies (similarly to multiple *pssA* copies) resulted in a nearly twofold increase of EPS synthesis and enhanced nodulation and symbiotic effectiveness [92]. Transcriptome profiling of the *rosR* mutant revealed the additional role of RosR in motility, synthesis of cell-surface components, carbon and nitrogen transport and metabolism, and other cellular processes [37]. Among a large group of genes (1106) differentially transcribed in the *rosR* mutant, a majority (63%) were up-regulated (suggesting that RosR functions mainly as a negative regulator), including *prsD*, *rapA1* (autoaggregation protein), *ndvA* (CG transport protein to the periplasm), and genes of various transcriptional regulators (i.e., nitrogen regulatory protein P-II, phosphate regulatory system PhoBR, LacI, LuxR-type regulator RaiR). In contrast, several genes involved in EPS synthesis (*pssA*, *pssC*, *pssI*, and *pssS*) and cell motility (including the regulator *rem* and several genes coding for flagellar proteins) were down-regulated in the *rosR* mutant. This finding suggests the participation of RosR in a complex regulatory network. Regulation of *rosR* expression is very complex and involves four regulatory proteins: RosR, CRP, NodD, and PhoB, which modulate *rosR* transcription in response to various environmental factors (phosphate limitation, flavonoids, carbon source) [81,82,93] (Figure 4). A high *rosR* expression is ensured by the action of its strong promoter and two additional (upstream promoter and TGN-extended −10) elements [82,94]. Moreover, other sequence motifs (RosR-box bound by RosR, a LysR motif recognized by NodD, cAMP-CRP, PHO boxes recognized by PhoB, and numerous inverted repeats important for *rosR* mRNA stability) are engaged in the regulation of *rosR* expression [81,82,93,94]. Regarding the effect of the C source, a slight repression of the *rosR* gene is observed in the presence of glucose but not in that of glycerol [81].

Some environmental factors (root exudates, phosphate limitation, and nitrogen starvation) positively affect the transcription of *pssA*, *pssB*, *pssO* and *pssP* [60,95]. Flavonoids and NodD enhance EPS synthesis in *R. leguminosarum* through the activation of *rosR* [93], suggesting linked positive regulation of the synthesis of both symbiotic signals (EPS and Nod factors) [96], as also happens in *S. meliloti*.

There are different regulatory proteins (ExoR, Psi, PsrA, ExpR, and PssB) that have a negative effect on EPS production in *R. leguminosarum*. ExoR is an orthologue of *S. meliloti* ExoR and is involved in *pssA* repression [44,58]. An *exoR* mutant produces 3-fold more EPS than the wild-type strain and induces the formation of both effective and ineffective nodules on the host plant [97]. The *pssB* gene is located upstream of *pssA* and negatively affects EPS production, although its precise role in this process remains to be established. *pssB* encodes an inositol monophosphate phosphatase, and mutants in this gene produce more EPS than the wild-type *R. leguminosarum* bv. *trifolii* and *viciae* strains and induce non-nitrogen-fixing nodules on their host plants [82,98,99,100]. The *psi* (a polysaccharide inhibition) and *psrA* (a polysaccharide restoration) genes have been found on the symbiotic plasmid (pSym) of *R. leguminosarum* bv. *phaseoli*, near the *nod-nif* region required for nodulation and nitrogen fixation [101]. Interestingly, these genes are absent in *R. leguminosarum* bv. *trifolii* and *viciae* genomes [83], indicating that some regulators of EPS production are biovar-specific. An interesting effect was observed for *psi*, *psrA*, and *pssA* genes. Multiple *psi* copies inhibited EPS synthesis and abolished *Phaseolus* nodulation, whereas additional copies of *psrA* or *pssA* overcame this inhibitory effect, indicating that balanced numbers of these genes are required for proper EPS synthesis [59,101,102,103].

EPS synthesis in *R. leguminosarum* is also influenced by quorum sensing (QS) (Figure 4). So far, four N-acyl homoserine lactone (AHL)-based QS systems have been identified (*cin*, *rai*, *rhi* and *tra*) in this rhizobial species; they are involved in several processes (e.g., plasmid transfer, nodulation, and EPS production) [104]. The *cin* system, encompassing the *cinR*, *cinI* and *cinS* genes, is at the top of this hierarchical regulatory network and controls the induction of the remaining QS (*rai*, *rhi*, and *tra*) systems. CinR is a transcription regulatory protein inducing *cinI* expression in response to a signal molecule AHL, whereas CinI is responsible for the synthesis of specific AHL (*N*-(3-hydroxy-7-*cis-*tetradecenoyl)-L-homoserine lactone). CinS is a small protein with an anti-repressor function that induces the expression of the two regulatory genes of the *rhi* and *rai* systems (*rhiR* and *raiR*) and also that of *plyB*, a gene coding for a glycosyl hydrolase that degrades HMM EPS [105,106]. Another protein, the LuxR-type regulator ExpR, which functions independently of AHL, is also involved in QS and EPS production (it induces expression of *raiR* and *plyB*). This protein is an orthologue of *S. meliloti* ExpR, although it regulates the expression of a lower gene number than *S. meliloti* (among them only one gene to with EPS synthesis, *plyB*) [105,106].

In *R. leguminosarum*, light negatively influences exopolysaccharide synthesis as well as attachment and nodulation. This environmental factor is recognized by a histidine kinase photoreceptor containing a light, oxygen, and voltage (LOV)-domain [41]. In the absence of light, as well as in a strain carrying an inactivated LOV-histidine kinase gene, the production of EPS is significantly higher than in the wild-type strain under light conditions. However, this regulatory pathway is not common, since this LOV-histidine kinase photoreceptor gene is present only in some genomes of *R. leguminosarum* bv. *viciae* and *trifolii* strains (e.g., Rlv3841, RtWSM2304 and RtWSM1325).

EPS plays an important role in the adaptation of *R. leguminosarum* cells to various environmental stresses in their free-living stage, providing protection against desiccation, heavy metals, salinity, and nutrient limitation [43,46,107]. This polymer is also essential for biofilm formation on both abiotic surfaces and plant roots, being a major component of the biofilm matrix. The presence of Zn^2+^ ions positively affects EPS production and biofilm formation by *R. leguminosarum* cells, although the exact mechanism is still unknown. EPS-deficient mutants are significantly more sensitive to these stress factors in comparison to the wild-type strains. Regarding symbiosis, EPS is crucial for effective interactions of *R. leguminosarum* bv. *trifolii* and *viciae* strains with their hosts. It is required for the initiation and propagation of long IT and protection against H_2_O_2_ generated by the host during bacterial invasion [46,108]. All mutants of *R. leguminosarum* bv. *trifolii* and *viciae* that do not produce EPS have an identical phenotype: on compatible host plant roots these mutants only induce the formation of small nodule-like structures devoid of rhizobia that are ineffective in nitrogen fixation, confirming that EPS is indispensable for the proper interactions of these rhizobia with their macrosymbionts [35,44,54,55,56,57,58,62,79]. On the contrary, the overproduction of EPS by *R. leguminosarum* bv. *trifolii* strains enhances symbiotic competitiveness and nodulation in clover [92].

## 4. *Sinorhizobium meliloti* 

*Sinorhizobium meliloti* is a rhizobial species characterized by its narrow-host range for nodulation and is restricted to a few legume genera such as *Trigonella*, *Medicago* and *Melilotus*, reviewed by [109]. Despite this fact, the high amount of tools and resources that have been generated along the last 20 years in the study of the interaction between *S. meliloti* and *Medicago truncatula* has made this interaction a model system for the study of rhizobia–legume symbiosis, particularly of the symbiotic interactions forming indeterminate nodules [10].

The genomic sequence of *S. meliloti* 1021 became available in 2001 [110] and shows that this genome is tripartite and composed of a 3.65-Mb chromosome and the 1.35-Mb pSymA and 1.68-Mb pSymB megaplasmids. Nodulation genes are harboured by the pSymA, whereas genes involved in the production of symbiotically relevant polysaccharides are distributed among the chromosome and the pSymB. The expression of *nod* genes is complexly regulated and includes three different copies of NodD (NodD1, NodD2, NodD3), and another LysR regulator, SyrM [111]. NodD1 and NodD2 are constitutively expressed and activate the expression of *nod* genes in the presence of flavonoids (such as luteolin or methoxychalcone) and plant betaines (such as trigonelline and stachydrine), respectively, although NodD2 can also interact with methoxychalcone. NodD3 induces *nod* genes in the absence of external compounds. However, NodD3 expression is activated by SyrM, and vice versa, thus forming a self-amplifying circuit. In addition to NodD3, SyrM also activates the expression of *syrA*, whose encoded product is a small protein located in the cytoplasmic membrane and that, lacking an apparent DNA binding domain, mediates the transcriptional up-regulation of genes involved in the biosynthesis of one of the two forms of EPS present in *S. meliloti*: the succinoglycan (or EPS I) [112]. Thus, *S. meliloti* NodD3 connects the production of two different symbiotic signals: Nod factors and EPS I. The global regulator NolR is not present in 1021, but in other *S. meliloti* strains, such as Rm41, it represses NodD1 and NodD2 and, thus, Nod factors production [113].

*S. meliloti* strains produce two different kinds of EPS (Figure 1) [114,115]: EPS I or succinoglycan, whose repeating unit is a octasaccharide composed of seven residues of D-glucose and one residue of D-galactose joined by β-1,3, β-1,4 and β-1,6 glycosidic linkages, and EPS II or galactoglucan, composed of disaccharide repeating units containing D-glucose and D-galactose in a molar ratio 1:1 linked by α-1,3 and β-1,3 bonds. In both cases there are non-sugar substitutions: acetyl, pyruvyl and succinyl groups in EPS I, whereas in EPS II most of the glucosyl residues are 6-O-acetylated and all the galactosyl residues are substituted with 4,6-O-pyruvyl groups. Both kind of EPS can be produced in HMM and LMM fractions [116], but only the LMM forms of these polysaccharides are symbiotically active, as will be discussed later. EPS I is constitutively produced by *S. meliloti* 1021 in a complex medium, whereas the production of EPS II is stimulated by phosphate-limiting conditions [117].

The production of both types of EPS relies on two different gene clusters that are harboured by the pSymB (Figure 2), reviewed by [20,114,115,118]. The biosynthesis of EPS I is directed by a 27 kb region composed of 19 *exo* and 9 *exs* genes and affected by different regulators, whose encoding genes are located on different replicons, including the *exo*/*exs* cluster. The gene cluster responsible for the production of EPS II is 30 kb long and is constituted by 30 *wgx* (formerly *exp*) genes. Both *exo*/*exs* and *wgx* genes code for the different proteins required for the production of precursors, the synthesis and decoration of the repeating units, polymerization, and transport to the external medium. The chromosomal *exoC* gene codes for a phosphoglucomutase that is also required for the conversion of glucose-6-phosphate into glucose-1-phosphate, a precursor of EPS I.

The regulation of the production of both *S. meliloti* EPS is extremely complex, taking place at the transcriptional, post-transcriptional, and post-translational levels, and is subjected to external signals, such as plant flavonoids, abiotic stresses, and the level of nutrients such as nitrogen and phosphate (Figure 4) [7,115,119,120]. The primary regulatory circuit controlling *exo* gene expression is called RSI (for ExoR, ExoS, ChvI) and is composed of three elements: ExoR, the histidine kinase ExoS, and the response regulator ChvI, which are located in the periplasm, the membrane, and the cytoplasm, respectively [121,122,123]. ExoS activates ChvI through phosphorylation, which, in turns, activate *exo* gene expression but also the chromosomal *exoR* gene, which codes for a repressor that physically interacts with ExoS, inhibiting the ExoS/ChvI two-component signalling and, thus, EPS production. This tripartite system is also involved in the regulation of flagellum biosynthesis genes [121], as well as in other important cellular processes, such as carbon source utilization [124]. ChvI, when phosphorylated, represses motility. The chromosomal gene *cbrA* also represses EPS production but enhances motility [125]. Similarly, the tripartite EmmA/EmmB/EmmC system, also encoded by chromosomal genes, increases motility and decreases EPS production [126]. Thus, in *S. meliloti* there are different pathways that regulate bacterial motility and EPS production in opposite ways. Another negative regulator of EPS I production is encoded by *exoX*, which belongs to the *exo*/*exs* cluster [127]. The ExoX protein is a small, inner membrane-attached protein that presumably interacts with and inhibits ExoY, the protein responsible for the addition of the first sugar residue (UDP-Gal) of the octasaccharide repeating unit to the lipid carrier located in the inner membrane. The *exsB* gene, also located in the *exo*/*exs* cluster, codes for another negative regulator of EPS I production, which, most probably, acts at the post-transcriptional level [115,128]. *S. meliloti* EPS I production is also negatively regulated by a Succinate-Mediated Catabolite Repression that operates in the presence of succinate (the preferred carbon and energy source for this bacterium) and involves the participation of a phosphotransferase system (PTS) and the HPr protein [129].

EPS I production is positively influenced by two other proteins that also act as positive regulators of the synthesis of another symbiotic signal, Nod factors. On the one hand, SyrM, as mentioned before, is a positive activator of NodD3 and, consequently, enhances Nod factor synthesis and also activates the synthesis of SyrA that, in turns, stimulates EPS I production through the RSI circuit [111]. On the other hand, the global regulator MucR regulates differentially the production of EPS I and EPS II through the transcriptional activation of *exo* genes and the repression of the *wgx* cluster. MucR also stimulates Nod factor production and represses bacterial motility and genes involved in nitrogen fixation, thus acting as an enhancer of the first steps of the nodulation process [96,130]. Another chromosomal gene, *exoD*, has also a positive effect on EPS I production, although the exact mechanism remains to be elucidated [127].

Phosphate (Pi) levels have a clear influence in the production of EPS II by *S. meliloti* [115]. Thus, EPS II production is repressed in the presence of high Pi concentrations (above 10 mM, found in nodules), whereas low Pi concentrations, such as those typically found in natural environments (0.1 to 10 µM), induces the production of this polysaccharide. This regulation is achieved through the two component system formed by the sensor kinase PhoR and the response regulator PhoB, which, under Pi limitation, induces the expression of the *wgx* transcriptional activator WggR (that also shows autoinduction activity), whereas that, as mentioned above, MucR represses the expression of *wgx* genes (including that of *wggR*) under Pi sufficiency [130].

In *S. meliloti*, the Sin QS system is also involved in the regulation of EPS production [115,131]. This QS system is composed of the LuxR-type transcriptional regulator SinR, and SinI, an autoinducer synthase responsible for the production of different long-chain AHL, and, in conjunction with another LuxR-type regulator called ExpR, regulates many different processes, such as the production of EPS I and EPS II, motility, chemotaxis, nitrogen fixation, and metal transport [96,132]. The expression of *sinI* is SinR-dependent and enhanced by AHL-ExpR. Regarding EPS production, ExpR, when activated by the AHL produced by SinI, induces the biosynthesis of EPS II but also the production of LMM forms of EPS I through the induction of *exsH*, which codes for an endo-1,3-1,4-β-glycanase that depolymerizes HMM succinoglycan to produce symbiotically active LMM succinoglycan [133,134]. It is important to remark that one of the most-studied *S. meliloti* strains, 1021, lacks this QS-mediated regulation of EPS production because this strain harbours a disrupted copy of *expR* [110]. Because of this, most of the studies carried out analysing the role of *S. meliloti* ExpR in EPS production have been carried out in strain Rm8530 [96].

Interestingly, in *S. meliloti* there is an inverse correlation between EPS production and motility functions reviewed by [115]: three different but linked regulatory systems that enhance EPS production (MucR, and the ExoR/ExoS/ChvI and ExpR/Sin regulatory systems) also suppress motility, showing that both processes are linked by global regulatory pathways. In *S. fredii* HH103, MucR1 also positively regulates EPS production and negatively genes involved in motility [87].

As in *R. leguminosarum* bv. *trifolii*, EPS and LPS production in *S. meliloti* are connected through the *exoB* gene, which codes for a UDP-glucose 4-epimerase that is responsible for the production of galactose, a sugar that is present in both polysaccharides [135].

The work carried out in late 80 s and early 90s by the German group headed by Dr. Pühler demonstrated the essentiality of *S. meliloti* EPS for the successful infection of *Medicago* roots and invasion of nodules [136,137,138]. Following Tn*5* mutagenesis, they identified different *S. meliloti* mutants that were unable to produce EPS and showed an Inf^-^phenotype with alfalfa. In addition, they identified many of the *exo* genes involved in EPS I production. In the model symbiosis *S. meliloti*/*M. sativa*, LMM forms of EPS are indispensable for a successful interaction, in such a way that the absence of EPS production leads to a lack of formation of IT, resulting in infection arrest [17]. In a very elegant work, it was showed that EPS I is more efficient than EPS II mediating both IT initiation and extension, and also that both kinds of EPS can be functionally replaced, although with lower efficiency, by another surface polysaccharide, KPS [139]. In fact, succinoglycan is the only form of *S. meliloti* EPS that can mediate the formation of IT on *M. truncatula*. KPS are typically formed by a disaccharide repeating unit, one of them being a 3-deoxy-D-manno-oct-2-ulosonic acid (Kdo) residue or a Kdo-derivative (Kdx). However, not all the *S. meliloti* strains produce a symbiotically active form of KPS, as in the case of *S. meliloti* 1021, which only contains a poly-Kdo that does not play any role in symbiosis [140]. Thus, this strain absolutely requires EPS, either succinoglycan or galactoglucan, for a successful interaction with alfalfa. In contrast, *S. meliloti* Rm41 *exo* mutants (such as AK631 and the *exoB* derivative of Rm41) are still able to induce the formation of nitrogen-fixing nodules on *Medicago* because this strain produces a biologically active KPS [8,141].

In addition to playing a key role in symbiotic signalling with the host plant, which most probably suppresses plant defence responses [142], a number of functional studies indicate that both *S. meliloti* EPS play a protective role against not only different abiotic stresses, such as detergents, salt, acidic pH, and heat [143] but also certain responses from the host plant, such as ROS that are present within IT [144] and, maybe, against antimicrobial peptides that are produced inside nodules of certain host legumes belonging to the IRLC (inverting-repeat lacking clade) such as *Medicago* [145]. *S. meliloti* EPS, especially EPS II, are also important for biofilm formation [146].

## 5. *Sinorhizobium fredii* 

The *Sinorhizobium fredii* species is constituted by fast-growing rhizobial strains with a broad host-range that usually, with exceptions such as NGR234, includes soybean (*Glycine max*) [147]. Among *S. fredii* soybean-nodulating strains, two groups can be distinguished for their ability (or not) to nodulate American commercial soybeans [148]. Thus, the most studied *S. fredii* strains vary in their symbiotic behaviour with soybeans. Strain NGR234 fails to nodulate both American and Asiatic soybeans [149], although recently it has been shown that this strain can effectively nodulate certain varieties of *Glycine soja*, the ancestor of soybeans [150]. Strains USDA257 and CCBAU 45436 [149,151] efficiently nodulated Asiatic varieties of soybean, such as Peking or Jing Dou 19, but failed to fix nitrogen on American soybeans such as Williams82. Strain HH103, instead, is able to induce the formation of nitrogen-fixing nodules on both Asiatic and American varieties of soybean [147]. The broad host range of HH103 and its ability to efficiently nodulate commercial varieties of soybean make this strain very attractive both for this study and its potential application as a legume inoculant. The HH103 complete genome sequence [152,153], as well as transcriptomic analyses of the effect of genistein in global gene expression in the wild-type strain and in mutants lacking the symbiotic regulators NodD1, TtsI, MucR1, or SyrM [87,154,155], have been published. These analyses have been complemented with studies on other symbiotic regulators, such as NolR or NodD2, as well as the analysis of the effect of inducer flavonoids on EPS production [156,157,158,159], and they will be commented on later.

Rodríguez-Navarro et al. [48] reported that the HH103 EPS structure is composed of D-glucose, D-galactose, D-GlcA, and pyruvic acid, in a 5:2:2:1 ratio and that it is partially acetylated (Figure 1). This structure is identical to that reported previously for strain NGR234 [49]. However, it is necessary to remark that an incorrect figure of the NGR234 EPS structure (in which the first GlcA residue is incorrectly linked to the main chain through a β-glycosidic linkage instead of the correct α one) has been published in later articles and reviews but was corrected in a recent review [8]. To our knowledge, *S. fredii* strains, in contrast to *S. meliloti*, produce one single type of EPS.

Genes responsible for EPS production in HH103 constitute a cluster located on the largest plasmid (pSfHH103e, 2096125-bp, Accession: NC_016815.1) and show the same organization (Figure 2) as that present in NGR234 [160]. This cluster includes genes involved in regulation (*exoX*), EPS polymerization and transport (*exoF*, *exoP*, *exoQ*, *exoK*), and structural genes coding for glucosyl or galactosyl transferases (*exoA*, *exoL*, *exoM*, *exoO*, *exoU*, *exoY*), an UDP-glucose 4-epimerase (*exoB*), an acetyl transferase (*exoZ*), and the protein responsible for the synthesis of UDP-glucose from glucose-1-phosphate (*exoN*). The *S. fredii* EPS and the *S. meliloti* EPS I are structurally very similar, but still there are some differences between them, such as the fact that the repeating unit is an octasaccharide in *S. meliloti* and a nonasaccharide in *S. fredii*, as well as the absence of succinyl decorations in *S. fredii* and GlcA residues in *S. meliloti*. Thus, although the organization of the genetic clusters responsible for the production of these two exopolysaccharides are pretty similar, there are 4 genes that are present in *S. meliloti* 1021 but absent in *S. fredii* strains: *exoV* (coding for a pyruvyl transferase), *exoW* (glycosyl transferase), *exoT* (Wzx-type transport protein), and *exoH* (succinyl transferase). The presence of GlcA in the *S. fredii* EPS is provided by the encoded product of *rkpK*, which belongs to the so-called *rkp-2* region, and codes for a UDP-glucose 6-dehydrogenase [161]. Interestingly, the absence of GlcA in a HH103 *rkpK* mutant, as previously described for an *R. leguminosarum exo5* derivative, results in the blocking of EPS molecule assembly and, thus, in an EPS-deficient phenotype [14,161]. The other gene that is present in the *rkp-2* region, *lpsL*, codes for a UDP-glucuronate 4-epimerase which catalyzes the conversion of GlcA into GalA. Both uronic acids are present in the *S. fredii* HH103 LPS [162], which justifies the fact that HH103 mutants in either *rkpK* or *lpsL* exhibited altered forms of LPS. As mentioned before, the *R. leguminosarum* bv. *trifolii* orthologue of *rkpK*, *exo5*, is also involved in CPS, EPS, and LPS production since GlcA is present in CPS and EPS, whereas GalA is present in LPS [14,47]. Interestingly, *S. meliloti* Sm21 also harbour the *rkp-2* region, since uronic acids are present in its LPS as well as, in contrast to HH103, in its K-antigen type capsular polysaccharide (KPS) [163]. However, the *S. meliloti rkp-2* region does not influence the production of EPS, since neither EPS I nor EPS II harbour uronic acids. Thus, the *rkpK* (or *exo5*) and *lpsL* genes constitute a really good example for illustrating how the same genes can be required for the production of different surface polysaccharides in different rhizobia, but also how interconnected can be the synthesis of these biomolecules (Figure 5). To the best of our knowledge, the *R. leguminosarum* gene (s) involved in UDP-GlcA conversion into UDP-GalA has not been described yet.

The regulation of EPS production in *S. fredii*, although it remains less studied than in the closely related species *S. meliloti*, has been shown to be directly connected to *nod* regulon (Figure 4). In *S. fredii* HH103, the presence of flavonoids able to induce *nod* genes led to a full repression of EPS production, as determined by NMR (nuclear magnetic resonance) experiments, this process being dependent on the symbiotic regulator NodD1, which positively regulates Nod factor production and the symbiotic type 3 secretion system, T3SS [159]. To our knowledge, this is the only case of EPS repression mediated by inducing flavonoids described in rhizobia so far. In fact, this kind of repression does not take place in other *S. fredii* strains where it has been analysed, such as NGR234 or USDA257 [159]. Previous works showed that, in *S. fredii* USDA193, genistein provoked changes in EPS production [164], but these changes were qualitative rather than quantitative, and it was not established whether they depend on NodD or not. Interestingly, the global repressor NolR, which negatively regulates the production of Nod factors and the symbiotic T3SS, is a positive regulator of EPS synthesis in HH103, since the lack of NolR results in a clear reduction of EPS production, which is almost complete even in the absence of flavonoids [157]. Thus, EPS production in *S. fredii* is connected to the *nod* regulon and modulated in a completely opposite way to that of Nod factors and the symbiotic T3SS. *q*PCR studies showed that two crucial genes for EPS production, *exoY2* (coding for the protein that should mediate the addition of the first sugar to the lipid carrier) and *exoK* (a glucanase that could be involved in the cleavage of HMM to LMM forms of EPS) are strongly repressed when genistein and NodD1 are present and also when *nolR* is inactivated [159]. Recent studies have shown that *S. fredii* SyrM, whose orthologue in *S. meliloti* is involved in the activation of EPS production, participates in this genistein-mediated repression, although the mechanism remains obscure [155]. In clear contrast, flavonoids enhance EPS production in *S*. *meliloti* 1021 in a process mediated by SyrM through the SyrA protein [111]. In *R. leguminosarum* bv. *trifolii* 24.2, the presence of flavonoids has also been shown to enhance EPS production, in this case through the RosR transcriptional regulator [93]. In *S. fredii* HH103 and CCBAU45536, MucR1, an orthologue of RosR, is also a positive regulator of EPS production [87,88], but its effect is not related to flavonoids and cannot counterpart the presence of this phenolic compounds in strain HH103. Transcriptomic studies performed in HH103 revealed that, among genes involved in EPS synthesis, MucR only affects the expression of an operon constituted by *exoY2*, *exoF1* (whose products are responsible for the addition of the first sugar residue, UDP-Gal, to the lipid carrier located in the plasmatic membrane) and *exoQ* (involved in EPS polymerization). Curiously, in the absence of flavonoids, the HH103 MucR1 protein is also an activator of different genes, most of unknown function, belonging to the *nod* regulon. This is a new indication of how different HH103 regulators (NolR, MucR1, SyrM) connect the production of EPS and the expression of at least part of the genes constituting the *nod* regulon. The production of *S. fredii* HH103 EPS is also somehow interconnected with that of CG [15,87]. Thus, the inactivation of the HH103 gene coding for the CG synthase (*cgs*) led to an increased transcription of *exoA* and the production of EPS, whereas the mutation of *exoA* not only resulted in the abolition of EPS synthesis but also in an increased production of external CG. The MucR1 regulator is clearly involved in this connection, since its inactivation negatively affects *exo* gene expression but also that of *ndvA*, which codes for the transporter involved in the transfer of CG from the cytoplasm to the periplasm [87]. 

In contrast to that found in *R. leguminosarum* and *S. meliloti*, *S. fredii* HH103 EPS production is not influenced by QS mechanisms since this strain lacks a functional copy of the *expR* gene [165]. However, the two QS systems identified in strain NGR234 have a positive effect on EPS production and repress genes involved in motility and chemotaxis [166].

The *S. fredii* NGR234 EPS has revealed to be indispensable for the formation of nitrogen-fixing nodules with several host-plants, such as *Albizia lebbeck* and *Leucaena leucocephala*, but the lack of this polysaccharide did not affect symbiosis with other hosts, such as cowpea (*Vigna unguiculata*). Interestingly, the inactivation of the NGR234 *exoK* gene, coding for a glycanase, resulted in a lack of production of EPS-derived oligosaccharides and in the same symbiotic phenotype as EPS^-^ mutants, indicating that, as described previously for *S. meliloti*, these NGR234 exo-oligosaccharides are crucial for symbiosis with certain host plants [167]. The symbiotic role of HH103 EPS has been studied in different host plants, either forming determinate or indeterminate nodules, and in no case has this polysaccharide has proven to be crucial. In determinate-nodule-forming legumes, the absence of HH103 EPS has different effects in symbiosis depending on the host legume: positive with soybean (slight increase in plant-top dry weight and significant increase in competitiveness ability), neutral with *Lotus burttii* (no differences in plant development) or negative with cowpea (slight decrease in plant-top dry weight and significant decrease in competitiveness ability) [48,168,169,170]. The fact that EPS appears to be slightly detrimental for the symbiosis of HH103 with its natural host, soybean, opens the question of whether EPS repression through *nod*-gene-inducing flavonoids is a mechanism for optimizing symbiosis with this legume. Regarding indeterminate-nodule-forming legumes, the HH103 *exoA* mutant induces the formation of nitrogen-fixing nodules with a very similar performance to those of the parental strain, on pigeon-pea (*Cajanus cajan*) and *Glycyrrhiza uralensis* [168,171]. The last legume belongs, as *Medicago*, to the IRLC clade [172]. In *S. meliloti*, EPS can be functionally replaced by KPS for effective symbiosis with alfalfa [139]. However, a double *exoA rkpH* mutant (producing no EPS nor KPS), although negatively affected when compared to HH103, is still able to induce the formation of nitrogen-fixing nodules on both pigeon pea and *G. uralensis* [168,171]. This fact is a good example of how “rules” (strict requirement of either EPS or KPS for effective symbiosis with an indeterminate nodule-forming legume) extracted for a model symbiotic system (*S. meliloti*/*alfalfa*) cannot always be applied to other symbiotic relations.

In free-living HH103 cells, EPS has proven to be essential for biofilm formation on abiotic surfaces [48,159], but it is not implied in attachment to soybean roots. In fact, and in contrast to *Bradyrhizobium diazoefficiens* USDA110, HH103 EPS (as in all *S. fredii* strains described so far) is unable to bind to soybean lectin. Regarding other physiological traits, the absence of EPS impairs the low surface motility exhibited by *S. fredii* HH103 and negatively affects tolerance to osmotic stress (either 50–100 mM NaCl or 5–10% sucrose *m*/*v*) but has no effect on sensitivity to the oxidizing agents H_2_O_2_ or paraquat [48].

## 6. *Mesorhizobium loti* 

The *Mesorhizobium loti* R7A and MAFF303099 strains, recently reclassified as *M. japonicum* [173], were isolated from *Lotus corniculatus* and *Lotus japonicus* nodules, respectively [174,175]. Both strains have been extensively studied since they are able to induce nitrogen-fixing nodules on the model legume plant *L. japonicus* Gifu [176,177]. The *M. loti* MAFF303099 and R7A genomes have been sequenced [178,179]; in both strains, the symbiotic genes are grouped in the symbiosis island, a mobilizable 500 kb DNA region able to make a non-symbiotic mesorhizobia into a *Lotus* symbiont. The symbiosis islands of these strains share a conserved backbone sequence of 248 kb with about 98% DNA sequence identity [180]. Interestingly, a comparison of the genomes of these two strains has shown that they have different secretion systems to introduce protein effectors into the plant cells. While a T4SS is present in R7A, the MAFF303099 strain possesses a T3SS, albeit both secretion systems seem to be involved in the determination of the host range and would be functionally exchangeable [180,181].

As has been described by Kelly et al. [50], *M. loti* R7A produces both HMM and LMM forms of EPS. The repeating subunit consists of a highly O-acetylated octosaccharide composed of D-glucose, D-galactose, D-GlcA and D-riburonic acid in a 5:1:1:1 ratio (Figure 1). The presence of the riburonic acid as a terminal residue has only previously been reported in *S. meliloti* IFO13336, but in the case of this strain, it is α-d-Rib and not β-d-Rib as occurs in R7A [21].

The genes responsible for EPS production in *M. loti* R7A and MAFF303099 are clustered in the chromosome (Figure 2), similarly as in *R. leguminosarum* strains. The cluster *exoQFYXUKTLAMONP* (from mlr5249 to mlr5276) genes is present in both strains, whereas other genes that may be involved in EPS biosynthesis, such as possible orthologues of *exoI* (mll0560, mll8119 and mlr0479), *exoZ* (mlr6758 and mlr8032) and *exoB* (mll7878) are dispersed in the genome [178]. Although the acetylation of the *M. loti* R7A EPS was found in three out of six possible sites, the gene responsible for this substitution remains to be elucidated since, as mentioned, there are two different genes showing high identity to the *S. meliloti* 1021 *exoZ* gene, mlr6758 and mlr8032 [21].

To our knowledge, specific studies designed to analyse the regulation of the *M. loti* EPS have not been published. In contrast, different studies have been focused on the symbiotic role of this polysaccharide in different interactions, such as those of *M. loti* NZP2037 with *Leucaena leucocephala* and *Lotus pedunculatus* [182], *M. loti* MAFF303099 with *L. japonicus* B-129 [183], and *M. loti* R7A with *L. corniculatus* and *L. japonicus* Gifu [50]. In the case of *M. loti* NZP2037, the symbiotic phenotype of EPS-deficient mutants depended on the host plant: these mutants induced effective nodules with *L. pedunculatus* (a determinate-nodule-forming legume) but empty nodules unable to fix nitrogen with *L. leucocephala* (which forms indeterminate nodules). In the case of *M. loti* MAFF303099, mutants unable to produce EPS showed the same symbiotic phenotype as the wild-type strain in symbiosis with *L. japonicus* B-129. The most recent studies regarding the symbiotic relevance of *M. loti* EPS have been carried out on the R7A strain. In that strain, the symbiotic response with *L. japonicus* Gifu and *L. corniculatus* plants differs depending on the EPS biosynthesis step affected by the gene mutation. While those mutants affected in the early steps of the EPS production (such as *exoA* and *exoB*) were able to form effective nodules (although in the case of the *exoA* mutant, the number of nitrogen-fixing nodules induced on *L. japonicus* Gifu was clearly reduced), the mutants affected in the middle to late steps of EPS synthesis were clearly impaired on both plants with more or less severity depending on the mutant tested (*exoO*, *exoU*, *exoK*, mlr5265 and mlr5266) [50]. Based on these results, the *M. loti* EPS would act as a positive signal to modulate the plant response and carry out a normal rhizobial invasion (IT formation and bacterial release). Bacterial mutants that are affected at the early steps of the EPS production lacked this polysaccharide but were able to avoid the defence responses and induce effective nodules although with a delay. The more severe symbiotic impairment phenotypes were shown by bacterial mutants producing a truncated EPS that may be recognised as a negative signal that would activate plant defence responses and, therefore, disrupt IT progression [50].

The molecular basis of the results commented above was supported by the recent finding of the *L. japonicus* Gifu transmembrane LysM receptor kinase, EPR3. This receptor is able to recognise and bind monomeric EPS molecules, thus monitoring the bacterial EPS status during the colonization and infection processes [18]. The recognition of the *M. loti* wild-type EPS by the EPR3 receptor acts as a positive signal allowing epidermal infection. However, the truncated EPS form (a penta-glycan) produced by a *M. loti exoU* mutant is recognised as a non-compatible EPS, and this negative signal results in the arrest of IT formation and the absence of the formation of nitrogen-fixing nodules. In the case of an *exoB* mutant that does not produce EPS, the absence of a negative signal allows infection and the development of nitrogen-fixing nodules but to a lower extent than in the case of the wild-type strain because of the lack of the positive signal (recognition of the appropriate EPS by EPR3).

Thus, the presence of this EPS receptor establishes a double checkpoint in the infection process of *M. loti* with *L. japonicus* Gifu plants. The *M. loti* R7A Nod factors are recognised by the NFR1 and NFR5 receptors and trigger the symbiotic signal transduction leading to the early steps of infection and nodule devolvement [184,185]. At the first steps of this signal cascade, the *epr3* gene is expressed, giving rise to EPS monitoring on the root infection zone and controlling the epidermal infection [18]. Recent results show that EPS3 is also expressed at the root cortex and nodule primordia, suggesting that EPS recognition is reiterated during symbiotic interaction, and it is also involved in intercellular cortical infection [19]. Recently, it has been shown that EPR3-like receptors, which present a structure that is quite different from that of LysM receptors involved in chitooligosaccharide (NF) perception, are ubiquitous in plants, suggesting that they might be surveillance receptors monitoring carbohydrates from different microbes associated with plant roots [186]. However, in the indeterminate-nodule-forming-legume *M. truncatula*, the orthologue of *eps3* (*Mtlyk10*), although playing a relevant role in symbiosis, appears to be uninvolved in the perception of the *S. meliloti* EPS I [187].

## 7. Conclusions and Perspectives

In this review we have tried to present the current knowledge about one of the most studied rhizobial surface polysaccharides, the EPS. Currently, we can point out the following relevant facts:(1)Rhizobial EPS production has been found to be extremely complex in all the rhizobial strains in which this issue has been addressed. Usually, it involves the participation of different regulatory systems, either positive or negative ones, including the participation of global regulatory proteins and, in some cases, that of QS systems.(2)In many rhizobia-legume symbioses, EPS is absolutely essential or very important for a successful interaction. This is especially true in symbioses forming indeterminate nodules, although the interactions between *S. fredii* HH103 and *G. uralensis* and *C. cajan* are a clear exception. Also, the fact that EPS plays a very important role in the success of rhizobial infection in the *M. loti*/*L. japonicus* couple, which forms determinate nodules, makes it difficult to formulate a general rule. Interestingly, EPS appears to be crucial for the symbiotic performance of several rhizobia exhibiting a narrow host range (such as *S. meliloti*, *R. leguminosarum*, and *M. loti*).(3)In some rhizobia, EPS production is connected with the *nod* regulon, but the type of connection (positive or negative) varies depending on the rhizobial strain. Thus, in *S. meliloti* and *R. leguminosarum* bv. *trifolii*, appropriate flavonoids induce Nod factor production and stimulates EPS biosynthesis, which is enhanced in parallel to that of Nod factors, whereas in *S. fredii* HH103, appropriate flavonoids induce Nod factor production but suppress EPS biosynthesis.(4)Rhizobial EPS, in addition to being crucial for biofilm formation and for passive protection against abiotic stresses, may play a crucial role as a molecular signal required for suppressing plant defence responses and, thus, allowing a successful infection process. There is clear evidence that, at least in *S. meliloti*, *R. leguminosarum* bv. *trifolii* and *S. fredii* NGR234, this signalling function is carried out by oligosaccharides derived from EPS. The recent discovery of the *Lotus japonicus* EPR3 receptor as a detector of the presence of the “correct” oligosaccharides for granting a successful infection process supports this idea. However, in this context, it is surprising that *S. fredii* HH103, a broad host-range rhizobial strain, represses the production of its EPS in the presence of any *nod* genes inducing flavonoids and, thus, in the early steps of its symbiotic interactions with any potential host legume. Might this repression be a strategy for nodulating potential host legumes that would stop infection when they perceive the HH103 EPS to inadequate? Clearly, more research is required to elucidate this issue.(5)Rhizobial EPS production may be interconnected with the biosynthesis of other symbiotic relevant surface polysaccharides. For example, in *R. leguminosarum* bv. *trifolii* and *S. fredii*, the synthesis of EPS and LPS are interconnected through the *rkp-2* region, whereas in *S. meliloti*, this genetic region participates in the production of KPS and LPS. At least in *S. fredii*, the production of EPS and extracellular CG is simultaneously controlled by the MucR1 regulator.(6)Rhizobial EPS symbiotic function can overlap with that of other surface polysaccharides, such KPS in the case of the interaction between *S. meliloti* and *Medicago*. However, this is not a general rule, as *S. fredii* mutants lacking EPS and KPS can still induce the formation of nitrogen-fixing nodules on *G. uralensis* and *C. cajan*.


In summary, EPS is a symbiotically relevant surface polysaccharide in most of the rhizobia–legume symbiotic interactions studied so far, and its production is complexly regulated together with that of other bacterial relevant characteristics, such as motility. However, the classical belief stating that EPS are specifically essential in interactions with indeterminate-nodule-forming legumes is not fully true, as demonstrated by the fact that they can also be highly important in determinate-nodule-forming interactions (such as *M. loti*/*L. japonicus*) and dispensable in the symbiosis between *S. fredii* and *G. uralensis*, which forms indeterminate nodules. Thus, in our opinion, the symbiotic importance of EPS cannot be predicted but must be analysed in every single rhizobial–legume interaction.

## Figures and Tables

**Figure 1 ijms-22-06233-f001:**
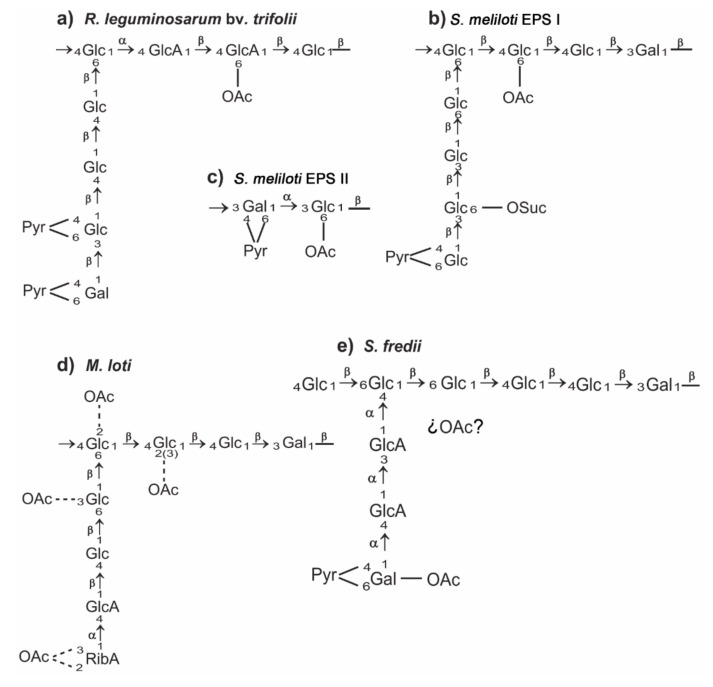
EPS structures of different rhizobia [7,8,16,21]. (**a**) *R. leguminosarum* bv. *trifolii*; (**b**) EPS I (succinoglycan) of *S. meliloti*; (**c**) EPS II (galactoglucan) of *S. meliloti*; (**d**) *M. loti*; (**e**) *S. fredii*. EPS structure is normally conserved at the species level but, in *R. leguminosarum*, it varies at the biovar level and in biovar *trifolii*, it may vary at the strain level. The structure shown in this figure is that found in most *R. leguminosarum* bv. *trifolii* strains, such as Rt42.2, RBL5599, and LPR5. Ac, acetate; Gal, D-galactose; GalA, D-galacturonic acid; Glc, D-glucose; GlcA, D-glucuronic acid; Pyr, pyruvate; RibA, D-riburonic acid; Suc, succinate.

**Figure 2 ijms-22-06233-f002:**
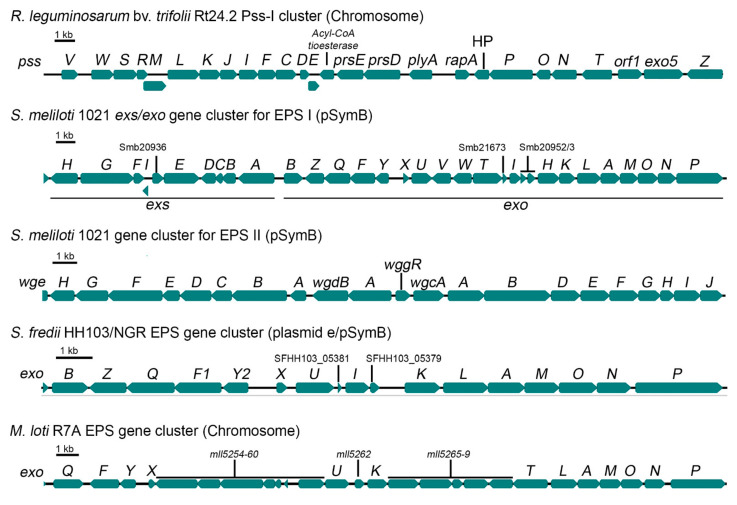
Genetic organization of EPS synthesis clusters found in *R. leguminosarum* bv. *trifolii* Rt24.2, *S. meliloti* 1021, *S. fredii* HH103 and NGR234, and *M. loti* R7A [30,35,48,49,50].

**Figure 3 ijms-22-06233-f003:**
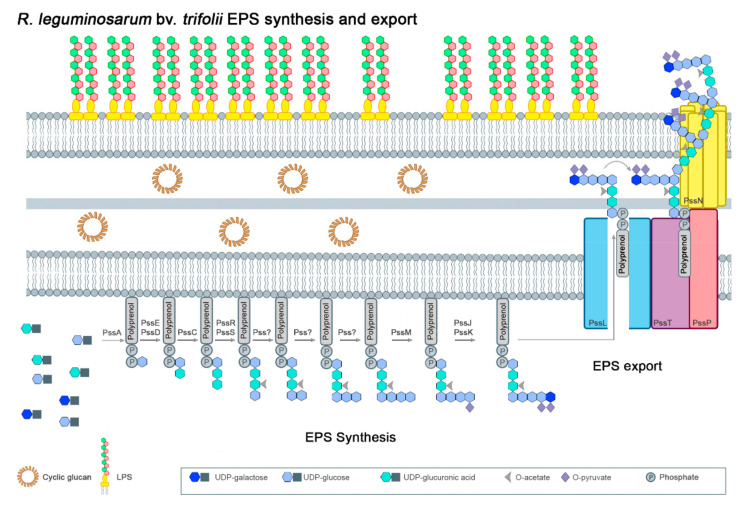
A model for EPS synthesis and export in *R. leguminosarum* bv. *trifolii* [29,35,36,53,55,62,63,64,65,66,67,68,69,70,71,72,73].

**Figure 4 ijms-22-06233-f004:**
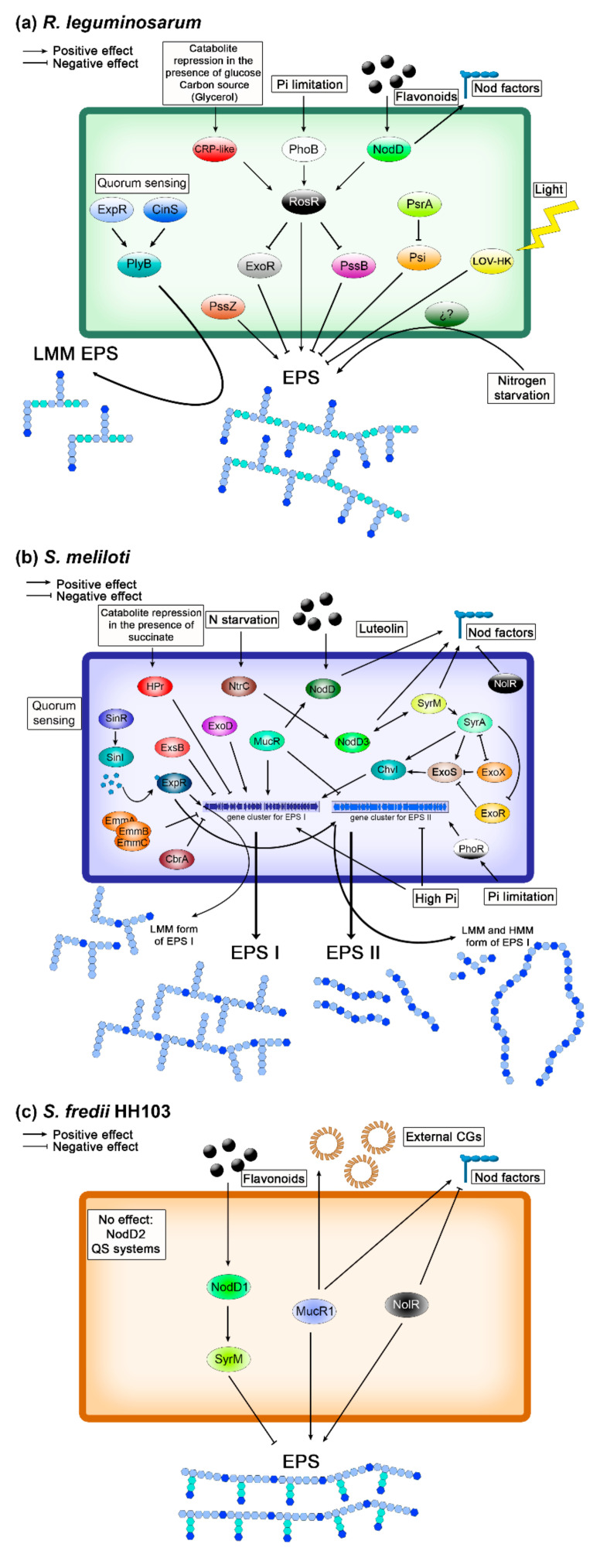
Regulation of EPS synthesis in different rhizobia. (**a**) *R. leguminosarum*; (**b**) *S. meliloti*; (**c**) *S. fredii* HH103. Details and references used are provided in the text. For each species, not all the regulatory proteins showed in the figure may be present in all strains. In *R. leguminosarum*, Lov-HK is present in some strains of bvs. *viciae* and *trifolii*, whereas *psi* and *psrA* are only present in bv. *phaseoli*. In *S. meliloti*, *expR* is disrupted in strain 1021, avoiding the regulation of EPS production by QS. In the case of *S. fredii*, the positive effect of NolR and the negative effect of NodD1 and flavonoids on EPS production have only been proven for strain HH103.

**Figure 5 ijms-22-06233-f005:**
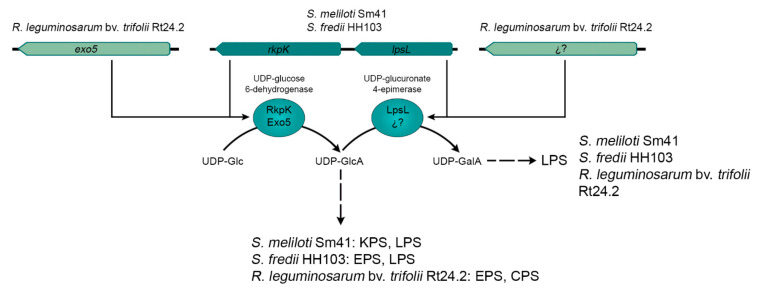
Genes involved in uronic acids production participate in the production of different surface polysaccharides depending on the rhizobial species [14,47,161,162,163].

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
