# Peer review of "Rhizobial Exopolysaccharides: Genetic Regulation of Their Synthesis and Relevance in Symbiosis with Legumes"

_ijms, 2021, doi:10.3390/ijms22126233_

Round 1

Reviewer 1 Report

 The topic of the review is quite relevant. To date, there is a lot of information about EPS (surface polysaccharides), their structure, genetic control, and functions in rhizobia. Therefore, the systematization of these data is of great importance. However, I think that this review is not ready for publication.

            There are many complaints about the text. There are repetitions, stylistic errors, often the text is too heavy. Information about the regulatory genes for the synthesis of EPS is not presented. It is confusing, that these genes are not shown in the figures.

            Some comments to figures: 1) Add references to figures, mark modifications. Figure 2 does not show all the genes mentioned, and their genomic localization is not always indicated. Regulatory genes are not shown in the figure.

            The role of EPS in symbiosis is not clearly shown. For example, the works of German scientists A. Puehler and A. Becker, who made a great contribution to the study of the genes for the synthesis of EPS of alfalfa nodule bacteria and their role in symbiosis, are not given.

           The conclusions are ambiguous. There is no comparative analysis of the genetic regulation of the synthesis of EPS and their role in the symbiosis between different types of rhizobia considered in this review. The importance of EPS for symbiosis remains unclear?

Specific comments.

77-79

In the introduction, the author gives the types of rhizobial surface polysaccharides, but gel-forming polysaccharide (GPS) and glucomannan are not mentioned (see reference 39, 58).

46..    Although there are exceptions (I would like a ref.), in most cases

Figures are uninformative, do not fully reflect the information available in the literature at the moment. In the captions to all figures references to the sources and modifications should be indicated.

Figure 2 shows not all 30 genes described in the text for one strain of R. leguminosarum bv.trifolii but only 26. For example, pssHG genes are located in the Pss-I cluster, their products are described, a role in the biosynthesis of a polysaccharide, but they are not shown in the figure. In the text, the author shows the important role in the biosynthesis of EPS of the pssA and rosP genes, which are not shown in the figure, and their localization in the bacterial genome is not indicated.

Why is only one Pss-I cluster presented?  

153.. In the case of R. leguminosarum, a great majority of genes engaged in EPS production are located on a large chromosomal region, called Pss-I [31-34] that encompasses nearly 30 genes and is highly conserved among all the so far sequenced R. leguminosarum genomes [28, 31,32,35-37], 

then the author writes about the importance of the rosR gene - 

264.. The global regulator RosR is involved in the positive regulation of EPS synthesis in R . leguminosarum. other genes the role of exoR, psi, psrA, expR, and pssB genes in negative regulation of EPS synthesis in R. leguminosarum has been confirmed.   

in the regulation of EPS synthesis, but these genes are also not shown in the figure.

The author presents the results for only one strain for each type of rhizobia, while several strains have already been described in the literature, it is desirable to compare the available data and compare their general and distinctive characteristics.

For S. meliloti, there are two clusters EPS1 and EPS2, the names of which do not appear in figure 2, indicate them, please. The number of exs genes is indicated as 2 (404..), but the figure shows more. Check to please the figure.

I recommend presenting the steps of the unit assembly… .with enzymes, genes, and references. There is a description in the text (pages 4-5)

203-206..  mutant nodules showed normal invasion and release of bacteria into plant cells,…

– change for in mutant nodules was shown normal invasion and release of bacteria into plant cells, …

There are repetitions of paragraphs (226-233 and 237-244), words (pp. 300-302 positively etc.)

282-286  Among a large group of genes (1106) differentially transcribed in the rosR mutant, a majority (63%) were up-regulated (suggesting that RosR functions mainly as a negative regulator), including prsD, rapA1 (autoaggregation protein ), ndvA (CG transport protein to the periplasm), and genes of various transcriptional regulators (ie, nitrogen regulatory protein P-II, phosphate regulatory system PhoBR, LacI, LuxR-type regulator RaiR). 

The word is missing in the sentence - genes of

297-299  Thus, slight repression of rosR expression is observed in the presence of glucose but not in that of glycerol [75]. 

The final sentence does not reflect the author's text on the regulatory function of the rosR gene in this paragraph 264-297.

308..  An exoR mutant produces 3-fold more EPS than the wild-type strain, and induces both types, ie, effective and non-infective nodules on the host plant [91]. 

Bad sentence style and instead of non-infective, to write - ineffective.

305, 338, 353 

often uses the word furthermore while the semantic meaning of this word is not clear. 

364..  Sinorhizobium meliloti is a rhizobial species characterized by its narrow-host range for nodulation, that is restricted to a few legume genera such as Trigonella, Medicago and Melilotus [reviewed by 103].

The author emphasizes the narrow specificity of alfalfa rhizobia, although they interact with three genera of leguminous plants and have effective cross-inoculation. At the same time, the narrow specificity of clover rhizobia is not mentioned in any way. And how does this fact affect the structure of the EPS and its function in symbiosis?

402..  In the description of EPS1 cluster, the author indicated 19 exo and 2 exs (a total of 21 exo / exs) genes, while reference 109 indicates that the EPS1 cluster contains 28 genes - who is right? Although the figure matches the figure from reference 109.

.

690.. small and non-infected nodules unable to fix nitrogen 

- does this mean tumors/ swell?

There is no reference to Fig. 2.for the genes of Mezorhizobium loti EPS synthesis.

745.. Conclusion 2 - In some rhizobia, EPS production is connected with the nod regulon, but the type of connection (positive or negative) varies depending on the rhizobial strain. Thus, in S. meliloti and R. leguminosarum bv. trifolii, EPS production is enhanced in parallel to that of Nod factors, whereas in S. fredii HH103, EPS and Nod factors are inversely regulated. 

The first sentence indicates strain dependence, while the next one compares different species.

770.. Conclusion 5 - Rhizobial EPS symbiotic function can overlap to that of other surface polysaccharides, such KPS in the case of the interaction between S. meliloti and Medicago. However, this is not a general rule, as S. fredii mutants lacking EPS and KPS can still induce the formation of nitrogen-fixing nodules on Glycyrrhiza uralensis, a legume belonging, as Medicago, to the IRLC clade. 

Where was it mentioned in the text?

Reviewer 2 Report

This work reports an extensive, updated and very complete review about symbiotic relevance and genetic regulation of rhizobial exopolysaccharides. It is worth publishing.  Only minor revision is considered.

In this work a significant number of genes involved in the EPS synthesis are mentioned. A table with genes and their function (based on mutant phenotypes) with different host plants would facilitate the review and summarize the results.

EPS function depends on the symbiotic couple analysed and I think it is important to mention the studied strain. In Fig. 1 four genera are mentioned but only in two are the name of the strains indicated. Figures 3 and 4 could also indicate the name of strains for which more information is available as in Fig.2.

In the Fig.2 , To standardize the name of the S. meliloti strain should be S. meliloti 1021, the same in line 594.

The text in the boxes in Fig. 3 are too small, as well as the text indicating "positive or negative effect".

I think M. japonicum R7A or M. japonicum MAF303099 should be used after line 655, in 658, 668, 674, 679, 686,691 etc.

Minor points

Line 136, “Lens” genus could be added to Pisum, Vicia and Lathyrus

There is a duplication of the text: Lines 237-244 are the same as 226-233.

Line 615, HH103

Line 638, pigeon-pea (Cajanus cajan).

Line 687, L. japonicus

Round 2

Reviewer 1 Report

The new version of this manuscript is much better and is almost ready for publication.

 But there are some small remarks.

325 line - slight repression of rosR expression – may be better changed on 

 - slight repression of rosR gene (we are not sure)

449 - circuit controlling exo gene expression is called RSI 

- explain RSI please

504 - Most of the studies carried out for analyzing the role of S. meliloti ExpR in EPS production have been carried out in strain Rm8530 – repetition!

521-522 - … very elegant work, it was showed that EPS I is more efficient than EPS II mediating both 522 IT initiation and extension,…  and   

523-525 - Thus, EPS-I appears to be the most efficient polysaccharide promoting a successful infection of Medicago by S. meliloti. - the same information!

516 – it is necessary to include some information from 132-134 ref. to the following text (518-531)

531-534 – In contrast, S. meliloti Rm41 exo mutants (such as AK631, and exoB derivative of Rm41) are still able to induce the formation of nitrogen-fixing nodules on Medicago since this strain produces a biologically active KPS [8,137]. – to correct this sentence because “strain AK631 carrying the exoB-631mutation is a Fix+ Exo− a derivative of S. meliloti Rm41 (Putnoky e.a.,1988, 1990)”, why are you using these ref-s 8, 137?

705-709 - In contrast to that found in R. leguminosarum and S. meliloti, S. fredii HH103 EPS production is not influenced by QS mechanisms since this strain lacks a functional copy of the expR gene [157163]. In contrast, the two QS systems identified in strain NGR234 have a positive effect on EPS production, meanwhile, they repress genes involved in motility and chemotaxis [158164]. – repetition! Unfortunately, I have not found information on the effect of QS on EPS production in the above text.

845-848 - Interestingly, EPS appears to be crucial for the symbiotic performance of several rhizobia exhibiting a narrow host range (such as S. meliloti, R. leguminosarum or M. loti), which might reflect a structural adaptation of their EPS for an adequate perception by the host. – A lot of speculations!!!

855-858- This fact, which does not take place in other S. fredii strains such as USDA257 or NGR234, suggests that HH103 EPS may be detrimental for symbiosis with some host plants. An HH103 EPS- mutant is more competitive for nodulating soybean than the wild-type strain. – This text would be better to include to the text  5. Sinorhizobium fredii. The special cases are not written in conclusions!

Author Response

IJMS-1229795, Cover letter of the 2nd revised version of the manuscript

Dear Dr. Xiaoshan Wan,

We have sent our revised version of manuscript ID IJMS-1229795 (“Rhizobial exopolysaccharides: genetic regulation of their synthesis and relevance in symbiosis with legumes”). We have also uploaded a corrected version of the Figure 2, since the original one contained a mistake on the location of exo genes of S. fredii.

We would like to sincerely thank again the two reviewers for their really useful comments and suggestions and for carrying out their work so quickly. As we mentioned in our last letter, they have contributed to improve the quality of the manuscript. These are our responses to the new suggestions and corrections of Reviewer 1:

The new version of this manuscript is much better and is almost ready for publication, but there are some small remarks.

1) 325 line - slight repression of rosR expression – may be better changed on  - slight repression of rosR gene (we are not sure)

Done.

2) 449 - circuit controlling exo gene expression is called RSI - explain RSI please

Done

3) 504 - Most of the studies carried out for analyzing the role of S. meliloti ExpR in EPS production have been carried out in strain Rm8530 – repetition!

The sentence has been corrected according to reviewer suggestion.

4) 521-522 - … very elegant work, it was showed that EPS I is more efficient than EPS II mediating both 522 IT initiation and extension,…  and  

Corrected.

5) 523-525 - Thus, EPS-I appears to be the most efficient polysaccharide promoting a successful infection of Medicago by S. meliloti. - the same information!

Reviewer is right. This sentence has been corrected.

6) 516 – it is necessary to include some information from 132-134 ref. to the following text (518-531)

Thank you for this suggestion. We have added some information coming from these references, stating the pioneer work of this group in the symbiotic relevance of S. meliloti EPS (lines 518-521).

7) 531-534 – In contrast, S. meliloti Rm41 exo mutants (such as AK631, and exoB derivative of Rm41) are still able to induce the formation of nitrogen-fixing nodules on Medicago since this strain produces a biologically active KPS [8,137]. – to correct this sentence because “strain AK631 carrying the exoB-631mutation is a Fix+ Exo− a derivative of S. meliloti Rm41 (Putnoky e.a.,1988, 1990)”, why are you using these ref-s 8, 137?

Reviewer is right. We have replaced reference 137 by Putnoky et al. 1990. We have also included ref. 8 because this issue is reviewed there.

8) 705-709 - In contrast to that found in R. leguminosarum and S. meliloti, S. fredii HH103 EPS production is not influenced by QS mechanisms since this strain lacks a functional copy of the expR gene [157163]. In contrast, the two QS systems identified in strain NGR234 have a positive effect on EPS production, meanwhile, they repress genes involved in motility and chemotaxis [158164]. – repetition! Unfortunately, I have not found information on the effect of QS on EPS production in the above text.

Corrected

9) 845-848 - Interestingly, EPS appears to be crucial for the symbiotic performance of several rhizobia exhibiting a narrow host range (such as S. meliloti, R. leguminosarum or M. loti), which might reflect a structural adaptation of their EPS for an adequate perception by the host. – A lot of speculations!!!

Reviewer is right. The last part of this sentence has been removed.

10) 855-858- This fact, which does not take place in other S. fredii strains such as USDA257 or NGR234, suggests that HH103 EPS may be detrimental for symbiosis with some host plants. An HH103 EPS- mutant is more competitive for nodulating soybean than the wild-type strain. – This text would be better to include to the text  5. Sinorhizobium fredii. The special cases are not written in conclusions!

Reviewer is right. In fact, this special case had been already mentioned in the S. fredii part, so we removed that sentence from the Conclusions.

We hope that the new version of the manuscript will be acceptable for publication in International Journal of Molecular Sciences. In addition to thanking to you for your work as Section Managing Editor, we would like to thank again the two Reviewers for their work on this manuscript.

Best regards,

Dr. José Mª Vinardell González

Full Professor

Department of Microbiology,

Faculty of Biology, University of Seville

Avda. Reina Mercedes 6, 41012-Sevilla (Spain)

+34 954554330

jvinar@us.es